# Mathematics teachers' reflective thinking: Level of understanding and implementation in their professional practices

**Abdulwali H. Aldahmash**[1]*, **Samar Ab. Alshalhoub**[2], **Majed A. Naji**[2]

**1** College of Education and Excellent Center for Science and Mathematics Education, King Saud University, Riyadh, Saudi Arabia, **2** College of Education, King Saud University, Riyadh, Saudi Arabia

* aaldahmash@ksu.edu.sa

**Data Availability Statement:** All relevant data are within the paper.

## Abstract

Mathematics teachers must be able to engage in self-reflection and think constructively about their professional activities to instruct students effectively. This is also important for students' ability to solve problems in class, as well as the challenges facing them and their societies in the future. This study aimed to examine in-service math teachers' reflective thinking skills in the context of how they are applied to their understanding of their teaching practices. The sample included 322 mathematics teachers, who responded to a questionnaire designed to elucidate how they were thinking reflectively about their teaching practice. The findings revealed that the teachers' practices concerning all the themes of reflective thinking addressed by the questionnaire, especially their ability to self-assess, fluctuated around the intermediate, and advanced levels. The implications of our findings are discussed in terms of extensive and continuous professional development centered on supporting ongoing growth in the reflective thinking skills of in-service math teachers.

## Introduction

Reflective practice [1–3] is described as a procedure that could help in smoothing teaching activities, and enhance students' learning and understanding. Therefore, it is considered as a major pushing wheel for teachers' professional development. During reflective thinking, teachers always reflect on and in their teaching practices [4], and their actions, which are considered as a means for their continuous professional development. When teachers undergo a thinking process to reflect on their teaching activities, they are considered reflective thinkers. Mathematics teachers' reflective thinking plays a major role in the realization of mathematics education objectives, the creation of solutions to problems facing mathematics teaching, and the evaluation of their experiences in the mathematics classrooms. Reflective thinking can also be considered an important research topic in terms of in-service and pre-service teachers' education. Efficient teachers are those who utilize reflective thinking in their teaching activities, as it is a necessary characteristic of successful and responsible teachers. Reflective teachers can contribute to the improvement of mathematics education through the production of highly qualified learners, who will be reflective thinkers and, thus, solvers of problems facing society in the

**Funding:** This research was supported by the Deanship of Scientific Research, Research at King Saud University, Group no. (RG-1440-123)

**Competing interests:** The authors have declared that no competing interests exist.

future [5]. Therefore, reflective thinking should be considered a major part of teacher education and their training programs.

Students have to be able to predict, seek simple formulas, and prove their ability to solve mathematics problems. Reflective thinking enables students (the ultimate "product" of the learning process) to assume responsibility for their learning, define their objectives, and make fruitful contributions to their learning processes [6]. Therefore, it is vital for teachers to develop students' reflective thinking abilities, which enables them to solve problems they consider difficult. However, if teachers do not possess these abilities themselves, they might not be able to assist their students appropriately. [7] indicated that a teacher's ability to reflect on his or her instructional activities is necessary for him or her to be effective in the classroom. The characteristic of self-reflection can help teachers cope with many of the problems that they face in their teaching practices as well as help them understand what their students are experiencing in their learning and comprehension of mathematics [6, 8–12]. For teachers to achieve a high level of self-reflection and reflective thinking [13], stressed that they need to collaborate with other teachers on their reflective practices.

[14] and [15] have also stressed the use of reflective thinking practices for the development of a teacher's professional growth and in-service professionalism. Currently, participation in society at large requires students and teachers to possess the ability to face various unfamiliar environments, experiences, and challenges. [16] indicated that this characteristic could be developed through understanding the systems and relationships that connect the world's entities, rather than viewing them as separate constituents. Teachers can achieve this by using teaching methods and strategies that align with the nature of mathematics [17, 18]. This approach may enable students to define mathematical problems better and discover innovative solutions [19]. Studies have indicated that this aim may not be achieved by teachers working individually; instead, its realization requires interdisciplinary content and collaboration (e.g., [20–23]).

It is necessary for teachers to develop the critical [24, 25] and reflective thinking skills required to teach mathematical and technical subjects. These skills can influence their ability to question and reflect on events and processes, in addition to allowing them to express various perspectives on current educational and scientific events [26]. As a result, training for such skills must be included in teachers' education and continuous professional development programs [27] to equip them with the ability to teach in different environments and contexts and help their students acquire those skills in turn [26, 28].

Reflective thinking skills are classified according to five core components, as follows: observation, communication, judgment, decision-making, and teamwork [29]. These five components might be essential, and all teaching activities should incorporate them. Teachers with a practice of engaging in dialog journaling, purposeful discussions, and teaching portfolios [30] can develop reflective thinking skills. [23] specified many reflective tools for supporting reflective thinking, including recording, writing, drawing, photography, learning journals, portfolios, lesson plans, co-teaching, collaborative practitioner inquiries, and action research.

Related studies have been done on teachers' reflective thinking on their practices. For example [29], proposed two instruments for the purpose of measuring teachers' reflective thinking skills. The two instruments are Closed-Ended Questionnaire and the Open-ended Questionnaire. They argue that the Open-Ended Questionnaire enables teachers to use their own words to explain the meaning of reflective thinking, as well as helps teachers identify the connection between the reflective thinking skills they have and the reflective thinking skills they need in their teaching. They also argued that the Open-ended Questionnaire could support teachers' deep understanding of the skills that they use in their teaching. In short, using these instruments might help researchers identify weaknesses and strengths in teachers' reflective thinking

and can pave the way to improving their reflective thinking. [8] found that assumption analysis, contextual awareness, imaginative speculation, and reflective skepticism are the important constituents of reflection. However, teachers neglect the practice of these four learning processes. This neglect makes them unable to perform deep reflection while practicing teaching. Similarly, [12], found that most teachers ignore reflection while practicing teaching, because they ignore those four learning processes. [29, 31], identified five important skills related to teachers' reflective thinking, which are; observation, communication, team working, judgment, and decision-making. In this study, a tool which includes four reflective thinking skills was used to measure teachers' reflective thinking about their practices, and that can be applied by teachers during mathematics lessons. These skills are; ability to self-assess, awareness of how one learns, developing lifelong learning skills, influence of beliefs about oneself, and self-efficacy [32].

## Problem statement

It is clear that there is a dearth of research regarding math teachers' reflective thinking regarding their teaching practices, especially in Saudi Arabia [30, 33]. By helping fill in this knowledge gap, this study hoped to strengthen the rationale for motivating further research exploring Saudi Arabian math teachers' reflective thinking regarding their teaching practices.

The research questions of this study are:

- Are math teachers practicing reflective thinking in their teaching profession?

- What do math teachers think about themselves and their teaching practices?

## Methods

The participants in this study sample were informed of the study objectives at the beginning, in the instruction part, of the questionnaire. They were also informed that participation was not obligatory, and that they did not have to write their names or identity. In addition, this study had been approved by the Research Ethics Committee of Deanship of Scientific Research, King Saud University.

The authors obtained informed written consent from the Research Ethics Committee of Deanship of Scientific Research, King Saud University via email.

I as a corresponding author ensure that all the authors mentioned in the manuscript have agreed for authorship, read and approved the manuscript, and given consent for submission and subsequent publication of the manuscript.

This study aimed to explore the reflective thinking skills of high school math teachers in Saudi Arabia. Data collection was conducted using a valid questionnaire including closed-ended and open-ended questions. Guided by the research questions, suitable statistical data analysis was performed to reach a more generalized understanding of the situation as well as the participants' trends regarding their teaching and professional practices for improving their reflective thinking skills [34].

## Sample

This study's main goal was to discover the levels of mathematics teachers' practices of reflective thinking. Therefore, during the first semester of the 2019/2020 academic year, the questionnaire was sent to the Makkah and Madinah districts and handed out to all secondary school mathematics teachers on duty at the time of distribution (N = 322) by the Excellent Center of Science and Mathematics Education, ECSME representatives. The participants were math

teachers with between 1 and 35 years of teaching experience, currently teaching in high schools in different districts in Saudi Arabia. All mathematics teachers who participated in the study sample teach in the same type of high schools, where students are aged 16 to 18 years.

## Instrument

There are many instruments that are used to measure reflective thinking skills. One of these instruments is a questionnaire developed by [8] called OPINIONNAIRE, accompanied by 5 open-ended questions. The other instrument is the Reflective Thinking Open-Ended Questionnaire, while the other is the Open-ended Questionnaire [35]. We used the Reflective Thinking Questionnaire constructed by [36], and modified by [8], followed by open-ended questions to help us get more insight into the perspective of reflective thinking. The open-ended questions lead teachers to use their own words to express the meaning of reflective thinking. It also allows us to know the reflective thinking skills that mathematics teachers need to be productive in their teaching, so that we can plan for specific professional development programs. The Open-ended Questionnaire might provide us with in-depth information about the situation regarding teachers' reflective thinking skills and their CPD needs in this field.

## Closed-ended questionnaire

This study used an instrument a questionnaire developed by [36] entitled "The Reflective Thinking of Teachers Questionnaire (RTTQ)" and boosted it with three open-ended questions. The close ended questionnaire included (33) items distributed among the following five themes:

- Ability to self-assess (12 items) with two sub-themes: "observing the performance" (9 items) and "making judgments" (3 items).

- Awareness of how one learns (9 items) with two sub-themes: "concepts and misconceptions" (3 items), "knowledge construction" (3 items), and "metacognition" (3 items).

- Developing lifelong learning skills (9 items) with three sub-themes: "developing my identity as a learner" (3 items), "transferring learning to other contexts" (3 items), and "understanding learning as a lifelong process" (3 items).

- Influence of beliefs about self and self-efficacy (3 items) with a single sub-theme: "developing a personal belief system."

The original questionnaire included four themes with (9) sub-themes and (33) items. The questionnaire included items inquiring about participants' demographic information. All the 33 items asked participants to rate each category using a five-point Likert scale (5 = strongly agree, 4 = agree, 3 = neutral, 2 = disagree, 1 = strongly disagree). The "neutral" response choice improved the survey's psychometric coherence, and, when the items were considered as a whole, had minimal effect on the overall reliability and validity of the instrument. The questionnaire topics were generated by [37], study, which was then modified by [8, 10], who targeted reflective thinking development. Each of the questionnaire items can be classified as representing an introductory (I), intermediate (In), or advanced (A) level of reflective thinking.

Ethics approval was obtained before the administration of the RTTQ questionnaire. The researchers informed participants about the purpose of the study and assured them that their information would be used only for this purpose. They were also informed that participation was voluntary and did not involve any compensation.

The minimum and the maximum length of the 5-point Likert scale were determined by calculating the range (5 − 1 = 4). The range was then divided by the greatest value of the scale (5) to get the least value (4 ÷ 5 = 0.80). The length of the cells was set as follows:

- Strongly agree: from 4.21 to 5.00.

- Agree: from 3.41 to 4.20.

- Neutral: from 2.61 to 3.40.

- Disagree: from 1.81 to 2.60.

- Strongly disagree: from 1 to 1.80.

## Open-ended questionnaire

The qualitative part included three open-ended questions aimed at obtaining deeper explanations of the quantitative results. [15], indicated that participants could express their opinions and reflect on their experiences through open-ended questions. The open-ended questions were as follows:

- Can you describe what you know about reflective thinking?

- Would you describe what you know about the importance of implementing reflective thinking in teaching practices?

- In your opinion, what are the characteristics of teachers who are reflective thinkers?

The answers obtained from the open-ended questions were coded using the process discussed by [38, 39], who described coding as "a word or short phrase that symbolically assigns a summative, salient, essence-capturing name to a portion of language-based or visual data" (p. 3). Those questions were originally in the Arabic language. The responses given by the participants were also in Arabic. They were analyzed and coded, and then the resulted themes, as well as the sample answers, were translated into English.

## Reliability and stability

The questionnaire was translated into the Arabic language as; the participants are Arabic speaking. It was translated back to the original language (English) to ensure the validity of the translation. The questionnaire was then submitted to a number of (8) experts in math education. As a result, no adjustments were made to the translation of the statements. To determine the stability coefficient of the tool, the researchers calculated the Cronbach's Alpha coefficient. The value of the coefficient of the reflective thinking tool was (0.873). This value revealed a positive indication of the use of the tool, and the reliability of the data collected through it.

## Findings

### First theme: The ability to self-assess

This theme includes four sub-themes. It shows that Table 1 shows the frequencies and percentages for the different levels of the theme related to the ability of math teachers to self-assess. The results of the first sub-theme, "observing the performance," "indicated that the percentages calculated from the teachers' answers for each of the five alternatives tended to be higher at the advanced level. This result was supported by the high mean value (4.60) recorded for the advanced level.

**Table 1. Descriptive statistics for mathematics teachers' ability to self-assess.**

| # | Items | Level | Frequencies of ratings | | | | | N | Weighted Mean |
|---|---|---|---|---|---|---|---|---|---|
| | | | 5 | 4 | 3 | 2 | 1 | | |
| *Observing the performance* | | | | | | | | | |
| 1 | I tend to follow orders rather innovate because I do not want to get in trouble. | I | 33 (10.4) | 97 (29.7) | 62 (19.6) | 90 (28.5) | 34 (10.8) | 316 | 3.02 |
| 2 | I try to think about what I teach my students in the area of expertise to enhance my lesson. | In | 113 (35.8) | 129 (40.8) | 37 (11.7) | 32 (10.1) | 5 (1.6) | 316 | 3.99 |
| 3 | I always think about what I have done during my lessons so that I can improve further. | A | 206 (64.2) | 108 (33.6) | 3 (0.9) | 3 (0.9) | 1 (0.3) | 321 | 4.60 |
| *Using feedback and evidence* | | | | | | | | | |
| 4 | I feel very anxious about the feedback my students give me; it is as though they are evaluating and judging me as a person. | I | 34 (10.8) | 71 (22.5) | 51 (16.1) | 113 (35.8) | 47 (14.9) | 316 | 2.78 |
| 5 | I think student feedback is important, as it helps me understand them better. | In | 102 (31.9) | 156 (48.8) | 42 (13.1) | 12 (3.8) | 8 (2.5) | 320 | 4.04 |
| 6 | I feel that student feedback is important as an indicator of my strengths and weaknesses. | A | 95 (29.7) | 163 (50.9) | 40 (12.5) | 20 (6.3) | 2 (0.6) | 320 | 4.03 |
| *Finding and analyzing patterns* | | | | | | | | | |
| 7 | I always think that what and how I did during my lesson is an important indicator of my effectiveness. | I | 153 (47.7) | 148 (46.1) | 15 (4.7) | 2 (0.6) | 3 (0.9) | 321 | 4.39 |
| 8 | I know that there are many areas, like content and context, that can make or break a lesson. | In | 124 (38.6) | 173 (53.9) | 17 (5.3) | 5 (1.6) | 2 (0.6) | 321 | 4.28 |
| 9 | I always try to look for areas of connectivity between what and how I teach using my life experiences. | A | 162 (50.8) | 138 (43.3) | 11 (3.4) | 3 (0.9) | 5 (1.6) | 319 | 4.41 |
| *Making judgments* | | | | | | | | | |
| 11 | I know I make mistakes but, sometimes, I feel that I cannot do anything about it. | I | 24 (7.6) | 73 (23.1) | 62 (19.6) | 113 (35.8) | 44 (13.9) | 316 | 2.75 |
| 12 | As a teacher, I know that the mistakes I make can influence the lives of my students. | In | 58 (18.3) | 142 (44.8) | 62 (19.6) | 46 (14.5) | 9 (2.8) | 317 | 3.61 |
| 10 | Whenever I am faced with a mistake that I have made, I try to make connections and learn from my experience and use it to move forward. | A | 171 (53.6) | 124 (38.9) | 13 (4.1) | 10 (3.1) | 1 (0.3) | 319 | 4.42 |

A = Advance = (4.37) In = Intermediate (3.98) I = Introductory = (3.24).

The results related to the second sub-theme, "using feedback and evidence," indicated that the teachers' ability to use feedback and evidence was most prevalent at the intermediate level, where the highest percentage of the respondents responded "strongly agree" or "agree" for both the intermediate and advanced levels. The mean scores also leaned toward the "agree" alternative for both the intermediate and advanced levels.

According to data in Table 1, mathematics teachers' ability to "observe their performance" was most prevalent at the advanced level, where the highest weighted average was 4.60. concerning the second sub-level, their ability to "use feedback and evidence" the level was also most prevalent at the advanced. Similarly, mathematics teachers ability to "find and analyze patterns" was also leaning towards the advanced level where the highest weighted average was 4.41. On the other hand, math teachers' ability to " *Using feedback and evidence* " was most prevalent at the intermediate level. The teachers' finally, mathematics teachers' ability to "make judgements" was most prevalent at the advanced level, where the highest weighted average was 4.42,–which means that teachers "strongly agreed" with this level. As a result, math teachers' ability to "self-assess" was found predominantly at the advanced level, where the value for the weighted average was 4.37.

**Table 2. Descriptive statistics for mathematics teachers' awareness of how one learns.**

| # | Items | Level | Frequencies of ratings (%) | | | | | N | Weighted means |
|---|---|---|---|---|---|---|---|---|---|
| | | | 5 | 4 | 3 | 2 | 1 | | |
| *Concepts and misconceptions* | | | | | | | | | |
| 13 | When students give me feedback, I do not take it too much into consideration because I feel that it is only their opinion. As long as I feel like I am doing my job, I do not worry about it. | I | 25 (7.89) | 55 (17.35) | 39 (12.30) | 147 (46.37) | 51 (16.09) | 317 | 2.55 |
| 14 | I think that it is important to consider students' feedback as it can help me improve what I am doing now so that I can perform better in the future. | In | 108 (34.07) | 162 (51.10) | 34 (10.73) | 10 (3.15) | 3 (0.95) | 317 | 4.14 |
| 15 | I like to take into consideration my past performance and integrate it with what I am doing in the present to help me better prepare for the future. | A | 139 (43.57) | 146 (45.77) | 20 (6.27) | 12 (3.76) | 2 (0.63) | 319 | 4.28 |
| *Knowledge construction* | | | | | | | | | |
| 16 | I am only interested in completing my assigned classes properly; basically, I do not have the time or interest to talk to my colleagues about their strategies and goals for their classes. | I | 33 (10.34) | 44 (13.79) | 34 (10.66) | 161 (50.47) | 47 (14.73) | 319 | 2.55 |
| 17 | I like to know how I am doing with my teaching, so, every opportunity I get, I want feedback from my supervisors, so that I can improve the way I deliver my lessons. | In | 138 (44.23) | 147 (47.12) | 18 (5.77) | 5 (1.60) | 4 (1.28) | 312 | 4.31 |
| 18 | Students today learn very differently from when I was in school. I need to look into new strategies to better deliver my lessons so that I can remain relevant now and in the future. | A | 171 (52.61) | 133 (41.69) | 13 (4.08) | 2 (0.63) | 0 (0.00) | 319 | 4.48 |
| *Metacognition* | | | | | | | | | |
| 19 | I have a certain way of delivering my lessons that I am comfortable with; I do not know why I do things the way I do—I just do. | I | 37 (11.78) | 119 (37.90) | 72 (22.93) | 80 (25.48) | 6 (1.91) | 314 | 3.32 |
| 20 | I am always interested in self-discovery so that I can apply knowledge about how I do things and hone myself to become a better teacher. | In | 154 (48.73) | 144 (45.57) | 14 (4.43) | 3 (0.95) | 1 (0.32) | 316 | 4.41 |
| 21 | I try to reflect on what I do during my lessons so that I can enrich my strategies with new and more effective ones. Sometimes, I can get inspiration by talking to my colleagues in other fields. | A | 142 (45.08) | 146 (46.35) | 23 (7.30) | 4 (1.27) | 0 (0.00) | 315 | 4.35 |

A = Advance = (4.37) In = Intermediate = (4.29) I = Introductory = (2.81).

## Second theme: Awareness of how one learns

This theme consisted of three sub-themes, as shown in Table 2. The results for the first sub-theme, "concepts and misconceptions," indicated that math teachers' awareness of concepts and misconceptions was most prominently observed between the intermediate and advanced levels, where the mean for the frequencies were 4.14 and 4.28, respectively. Meanwhile, the results of the second sub-theme, "knowledge construction," indicated that the teachers' ability to construct knowledge also concentrated around the intermediate and advanced levels. Similarly, the results for the third sub-theme, "metacognition," showed that teachers' belief in their metacognitive ability was most prevalent at the intermediate and advanced levels; the intermediate weighted average for the frequency of this level was 4.03.

The means for the categories of the theme, "awareness of how one learns," indicated that teachers' abilities to identify "concepts and misconceptions," "knowledge construction," and "metacognition" are predominantly at the advanced level, where the highest mean was found to be for this level (4.37). however, the percentages for the other levels still significantly high, which may indicate that math teachers need training on the reflective thinking and its use in teaching profession. we conclude that an advanced level predominance in theme "teachers' awareness of how one learns show that math teachers are, to some extent, aware of how their students learn and then might be able to teach students properly.

**Table 3. Descriptive statistics for mathematics teachers' developing lifelong learning skills.**

| # | Items | Level | Frequencies of ratings | | | | | N | Weighted Mean |
|---|-------|-------|------|------|------|------|------|---|------|
| | | | 5 | 4 | 3 | 2 | 1 | | |
| *Developing an identity as a learner* | | | | | | | | | |
| 22 | Sometimes, the feedback I get from my students and supervisor is so confusing that I do not know what to make of them. I do not think it is going to help me learn anything about the way I conduct my lessons. | I | 18 (5.75) | 53 (16.93) | 82 (26.20) | 127 (40.58) | 33 (1054) | 313 | 2.67 |
| 23 | I know I am still learning to be a better teacher, and the feedback I get from students and supervisors can help improve my future performance | In | 146 (45.77) | 136 (42.63) | 27 (8.46) | 6 (1.88) | 4 (1.25) | 319 | 4.30 |
| 24 | I know that I am learning about my profession all the time and I already have a set of practices that I am comfortable with; although, the feedback I get from students, and my supervisor can help me improve these practices even more. | A | 140 (43.89) | 162 (50.78) | 13 (4.08) | 4 (1.25) | 0 (0.00) | 319 | 4.37 |
| *Transferring learning to other contexts* | | | | | | | | | |
| 25 | Generally, I receive positive comments from students, so I think that I am doing quite well as a teacher overall. | I | 93 (29.15) | 166 (52.04) | 44 (13.79) | 12 (3.76) | 4 (1.25) | 319 | 4.04 |
| 26 | I know that feedback is just others' opinions about me. There must be some truth in what they see; if there wasn't, they would not make these remarks. I need to weigh the feedback I get against some of the opinions and assumptions I have about myself. | In | 92 (28.75) | 201 (62.81) | 21 (6.56) | 2 (0.63) | 4 (1.25) | 320 | 4.17 |
| 27 | I know I make assumptions about many things, and when others give me their opinions about how I am teaching, I must put them into perspective. After all, I can learn from the feedback I receive. | A | 77 (24.06) | 208 (65.00) | 28 (8.75) | 7 (2.19) | 0 (0.00) | 320 | 4.11 |
| *Understanding learning as a lifelong process* | | | | | | | | | |
| 28 | I know what I am doing as a teacher and do not spend much time reflecting on my teaching practices. | I | 34 (10.73) | 81 (25.55) | 61 (19.24) | 123 (38.80) | 18 (5.68) | 317 | 2.97 |
| 29 | I know I have my strengths and weaknesses and that teaching is a difficult job. I need to constantly look at my practices to be more effective with my lessons. | In | 123 (38.56) | 172 (53.92) | 21 (6.58) | 2 (0.63) | 1 (0.31) | 319 | 4.30 |
| 30 | I know how I present my classes influences how my students will behave toward the subject. Every time I present a class, I need to be cognizant that I should reflect on how I have taught and make changes next time if necessary. | A | 105 (32.92) | 187 (58.62) | 27 (8.46) | 0 (0.00) | 0 (0.00) | 319 | 4.24 |

A = Advance = (4.24) In = Intermediate = (4.26) I = Introductory = (2.23).

## Third theme: Developing lifelong learning skills

As shown in Table 3, the theme related to teachers' ability to develop lifelong learning skills consisted of three sub-themes. The results for the first sub-theme, which was related to teachers' ability to "develop an identity as a learner," were predominantly between the intermediate and advanced levels, but it was leaning towards the Advanced level, where the highest mean for the frequencies of this level was 4.37. The results for the math teachers' ability to "transfer learning to other contexts" showed fluctuation among the introductory, intermediate, and advanced levels, but the highest mean was for the intermediate level (X = 4.17). Similarly, teachers' ability to "understand learning as a lifelong process" was found to lean towards the intermediate level, where the highest weighted average was 4.30. In short, mathematics teachers' developing lifelong learning skills was found to be prominent at the intermediate level, because the highest mean was 4.26 belongs to this level. This indicated that teachers' need to improve their abilities to identify "concepts and misconceptions," "knowledge construction," and "metacognition" are predominantly at the [intermediate level and, hence their ability to developing lifelong learning skills is intermediate, Where the highest mean was for the intermediate level. This could mean that teachers are in need for training in reflective thinking especially in this theme.

**Table 4. Descriptive statistics for math teachers' views about the influence of belief about self and self-efficacy.**

| # | Items | Level | Frequencies of ratings | | | | | N | Weighted mean |
|---|-------|-------|---|---|---|---|---|---|---|
| | | | 5 | 4 | 3 | 2 | 1 | | |
| *Developing a personal belief system* | | | | | | | | | |
| 31 | I believe that I need to take care of my needs before I can take care of other people's. | I | 59 (18.50) | 120 (37.62) | 82 (25.71) | 51 (15.99) | 7 (2.19) | 319 | 3.54 |
| 32 | I know that what I believe about myself and others will ultimately control my behaviour. | In | 55 (17.19) | 123 (38.44) | 65 (20.31) | 65 (20.31) | 12 (3.75) | 320 | 3.45 |
| 33 | I am aware of my beliefs and know that these beliefs influence my behaviour toward myself and others. | A | 65 (20.50) | 152 (47.95) | 50 (15.77) | 41 (12.93) | 9 (2.84) | 317 | 3.70 |

## Fourth theme: Influence of belief about self and self-efficacy

The results of the fourth theme, "developing a personal belief system,", which consist only of one subtheme "*Developing a personal belief system*" as appear in Table 4 indicated that math teachers' beliefs about their ability to "develop a personal belief system" seams at the advance level. However, all the values of the means were found to be centered around the "agree" alternative in which the grading rules was are located between 3.45 and 3.70. however, the highest mean was for the advanced level, indicating that their ability to develop personal belief system and then their views about the influence of belief about self and self-efficacy is leaning towards the advanced level. This may indicates that their ability to think reflectively about the "influence of belief about self and self-efficacy" are not properly well established, which show that they need much training in reflective thing as a whole and this theme in particular.

## Summation of mean and percentages for each level and sub-level of reflective thinking abilities and the whole scale

Table 5 includes the weighted means and the weighted percentages for mathematics teachers' responses to all four themes included in the instrument. The results show that level (A) is

**Table 5. Mean results for each of the three levels for the four skills and the whole scale.**

| Theme | Sub-theme | Levels | | | | | |
|-------|-----------|--------|---|---|---|---|---|
| | | A | | In | | I | |
| | | $\bar{X}$ | % | $\bar{X}$ | % | $\bar{X}$ | % |
| **Ability to self-assess** | Observing the performance | 4.60 | 92% | 3.99 | 79.8 | 3.02 | 60.4 |
| | Using feedback and evidence | 4.03 | 80.6 | 4.04 | 80.8 | 2.78 | 55.6 |
| | Finding and analysing patterns | 4.41 | 88.2 | 4.28 | 85.6 | 4.39 | 87.8 |
| | Making judgments | 4.42 | 88.4 | 3.61 | 72.2 | 2.75 | 55 |
| **Total averages of the theme** | | 4.36 | 87.2 | 3.98 | 79.6 | 3.23 | 64.6 |
| **Awareness of how one learns** | Concepts and misconceptions | 4.28 | 85.6 | 4.14 | 82.8 | 2.55 | 51 |
| | Knowledge construction | 4.48 | 89.6 | 4.31 | 86.2 | 2.55 | 51 |
| | Metacognition | 4.35 | 87 | 4.41 | 88.2 | 3.32 | 66.4 |
| **Total averages of the theme** | | 4.37 | 87.4 | 4.29 | 85.8 | 2.80 | 56 |
| **Developing lifelong learning skills** | Developing an identity as a learner | 4.37 | 87.4 | 4.30 | 86 | 2.67 | 53.4 |
| | Transferring learning to other contexts | 4.11 | 82.2 | 4.17 | 83.4 | 4.04 | 80.8 |
| | Understanding learning as a lifelong process | 4.24 | 84.8 | 4.30 | 86 | 2.97 | 59.4 |
| **Total averages of the theme** | | 4.24 | 84.8 | 4.25 | 85 | 3.24 | 64.8 |
| **Views about the influence of belief about self and self-efficacy** | Developing a personal belief system | 3.70 | 74 | 3.45 | 69 | 3.54 | 70.8 |
| **Total averages of the tool** | | 4.17 | 83.43 | 3.99 | 79.8 | 3.20 | 64 |

**Table 6. Effect of experience on math teachers reflective thinking.**

| level | Experience | Mean | DF | F | Sig. |
|---|---|---|---|---|---|
| A | 1 | 4.3508 | 299 | 6.675 | .001 |
| | 2 | 4.1315 | | | |
| | 3 | 4.1284 | | | |
| In | 1 | 4.0755 | 299 | 1.121 | .327 |
| | 2 | 3.9750 | | | |
| | 3 | 3.9834 | | | |
| I | 1 | 3.0914 | 299 | 1.467 | .232 |
| | 2 | 3.2574 | | | |
| | 3 | 3.2041 | | | |

predominant over the other two levels. The results showed also that the intermediate level (In) leads the introductory level (I). However, it is clear that all levels scored high, which may indicate that high percentage of mathematics teachers whose scores fall in the intermediate or the introductory levels need training in issues and skills related to the use of reflective thinking in their teaching practices.

## Effect of experience of teachers reflective thinking

According to Table 6, it is clear that the teachers' level of reflective thinking fluctuated among the three levels of thinking assigned by the instrument developers. However, the results showed that the advanced level (A) dominated the intermediate level (In), which in turn dominated the introductory level (I). Regarding the effect of experience on mathematics teachers' reflective thinking on and in their teaching, it seems that it has a very small impact, except for level A (Advanced), where short experienced teachers scored higher than medium and short-experienced teachers. It is clear that there are significant differences between short experience and both medium and long experience in favour of short experience.

## The qualitative part

To deeply understand the results evolved from the quantitative part, we used open-ended in combination with the close-ended questionnaire. Three open-ended questions were used. The first question was, *"Can you describe what you know about reflective thinking?"* Concerning Math teachers' answers to this question, many of the teachers were found to hold vague correct perceptions of the concept of reflective thinking. For example, some teachers describe reflective thinking as thinking deeply about teaching processes. Others described reflective thinking as pausing at each step while teaching. However, some other teachers had some ideas about reflective thinking. For example, one secondary male math teacher said, *"every math teacher has to think reflectively about each step of his teaching ahead of conducting that step"*. By looking at all the teachers' answers, one could conclude that most of the participating mathematics teachers did not have a clear view of reflective thinking. They did not regard it as one of the patterns of thinking in general, and that it shares skills for this type of thinking, especially skills related to teaching mathematics. Mathematics teachers also lack knowledge of the tools that can be used to determine their level of reflective thinking. Many of the teachers did not have any idea about the students' need for reflective thinking.

The teachers' answers to the second open-ended question: ***Would you describe what you know about the importance of implementing reflective thinking in teaching practices?*** Their answers to this question were related to their answers to the first question in the interview,

where some teachers were able to identify some characteristics of a teacher practicing reflective thinking. For example, a secondary male teacher quoted *"math teachers who have strong reflective thinking should have deep mathematics knowledge. They should pause ahead of each step of teaching, armed with patience, keen in their evaluation"*. Some other teachers provided other characteristics, such as; having wealth of knowledge about students and their individual differences, so that they could take into account these differences in planning for their teaching. Other teachers indicated that a teacher who practices reflective thinking tends to discuss his/her teaching experiences with fellow teachers, supervisors, and trainers, and is keen on developing teaching practices continuously.

For this study, the themes of the tool used in this study represent the characteristics of reflective teachers, including math teachers:

- *Ability to self-assess; "observing the performance" and "making judgments" (3 items).*

- *Awareness of how one learns; "concepts and misconceptions", "knowledge construction", and "metacognition".*

- *Developing lifelong learning skills; "developing my identity as a learner", "transferring learning to other contexts" and "understanding learning as a lifelong process".*

- *Influence of beliefs about self and self-efficacy; "developing a personal belief system."*

- *Teacher reflection as retrospective analysis (ability to self-assess).*

Furthermore, most of the teachers did not provide answers to the third open-ended question, "***In your opinion, what are the characteristics of teachers who are reflective thinkers?***
Some responses consisted of ambiguous statements. For example:

*A male secondary science teacher (#3) wrote what it means "they are slow teachers because they spend most of their time thinking"*;

*A female science teacher (# 44) wrote, "They are better teachers."*

On the other hand, some teachers gave some answers; for example;

*One female science teacher (# 51) expressed that, "they always think about each step of their teaching, they always think carefully about the outcomes when they prepare their lesson, and they always revise their teaching strategies."*

## Discussion

In the discussion part, we will first discuss the quantitative results, and then follow that with the presentation and the qualitative part and the discussion. The results of the first theme categories (or sub-themes) indicated that math teachers' ability to "observe their own performance" was observed at the "advanced" level, which meant that they were "always thinking about what they had done during the lessons so that they could improve more. Their ability to "use feedback and evidence" was observed at both the intermediate and advanced levels, while their abilities to "find and analyze patterns" and "make judgments" were observed at the advanced level. This finding means that teachers "think students' feedback is important, as it will help them understand their students better," and "feel that students' feedback is important, as it will give them an indicator of their strengths and weaknesses." However, the reflective thinking ability of a considerable percentage of teachers was also observed at the intermediate

level and introductory levels. In other words, it might be concluded that an advanced level predominance in theme "ability to self-assess proof that math teachers are, to some extent, aware of how their students learn and then might be able to teach students properly and to self-assess their work in the classrooms. However, the results concerning the magnitude of intermediate and introductory levels could indicate that math teachers are still in need of intensive programs involving the four skills associated with this theme to enable them to reflect on their behavior in the classroom, especially in the area of self-assessment of their work Various studies (e.g., [9, 37]) support this conclusion and have indicated that this ability is vital for teachers. They have argued that a teacher's ability to self-assess "takes reflection as bending thoughts to incorporate prior experiences" and stressed that such experiences might have a noticeable impact on teachers' practices. They also indicated that reflective analysis according to a model for the development of reflective thinking should take the ability to self-assess into consideration.

The results for the second theme, "awareness of how one learns," as already demonstrated by the values of the means, indicated that teachers' "abilities to identify concepts and misconceptions," knowledge construction, and "metacognition" were found predominately at the advanced level. The level of the theme as a whole was found to fluctuate between the intermediate and advanced levels, but it leans towards the advanced level. However, a significant percentage of the teachers were found to think reflectively at the intermediate and introductory levels, indicating that teachers need assistance in improving their awareness of how people learn. This knowledge can ensure success in their teaching and help students learn better. It may also impact on teachers' learning of pedagogical and scientific concepts. In this regard [40], proved that pre-service teachers should have in-depth mathematical knowledge related to the topic, which could be achieved if they grasped the learning processes. Otherwise, they might not be able to "regulate their activities or make proper instructional explanations, thus conducting teacher-centered lessons as a way out."

Concerning the third theme, "developing lifelong learning skills," the means for the categories indicated that teachers' abilities to identify "Developing an identity as a learner," "Transferring learning to other contexts" and "Understanding learning as a lifelong process" also fluctuated between the introductory, intermediate and advanced levels. However, mean value of the whole theme are leaning towards the intermediate level. This finding indicated that teachers moderately "know that they are learning about their profession all the time and already have a set of practices that they are comfortable with, although the feedback they get from students and their supervisors will help them improve those practices even more." They also know that they make assumptions about many things and "when others give them their opinions about how they are teaching, they must put it into perspective. After all, they can learn from all the feedback they get. " Finally, teachers indicated that they moderately know how they "present their classes in such a way that influences how their students behave toward the subject. Every time they present a class, they need to be cognizant that they need to reflect on how they have taught and make changes the next time if necessary. The collective results of the whole theme indicate that teachers' need to improve their abilities to identify "concepts and misconceptions," "knowledge construction," and "metacognition" are predominantly at the [intermediate level and, hence their ability to developing lifelong learning skills is intermediate. This could mean that teachers are in need for training in reflective thinking especially in this theme. " In this regard [41], asserted that developing lifelong skills is essential for teachers to teach effectively, and cultivating lifelong learning is considered an important requirement for teachers in the 21st century.

The results for the theme "developing a personal belief system" indicated that math teachers' beliefs about their ability to develop a "personal belief" were not very clear. The mean value

appears to fall in the advanced level. however, all the values of the means were found to center around the "agree" alternative for all three levels of reflective thinking, which meant that the teachers' ability to think reflectively about the "influence of belief about self and self-efficacy" are fluctuated among the introductory, intermediate, and advanced levels, demonstrating that they are critically in need of developing this skill. Studies [32, 42, 43] have indicated that self-efficacy is necessary for developing teachers' collective efficacy, which may have a positive effect on their job satisfaction and emotions. Therefore, to be able to teach effectively, there is a great need to furnish in-service teachers with continuous professional development (CPD) programs that promote their self-efficacy and beliefs about the teaching profession.

Finally, [8, 44], indicated that students who have strong reflective thinking abilities know much about what they learned and are able to access and control what they know, need to know, and, as a result, can bridge that gap during learning situations. According to [4, 46], this is happening because of the presence of a wide range of thinking skills involved in critical thinking, which enable them to integrate these skills by making judgments about any evolving situations. In addition, reflective thinking may prompt the thinker to reach a reasonable solution to complex problem-solving situations by providing learners with the chance to use the best set of strategies to achieve their goals. From his side [4, 45], reasoned that reflectively practicing teachers are more likely to adopt strategies that help students be good reflective thinkers, a step that may facilitate critical thinking among students.

Regarding the effect of experience on mathematics teachers' reflective thinking on and in their teaching, it seems it has very small impact, except for level A (Advanced), where the short experienced teachers scored higher than medium and short experienced teachers. The predominance of the levels "Advanced, intermediate and Introductory" in the light of Tables 1–3, could be justified by suggesting that mathematics teachers graduated from different institution around the world. This feature might have been impeded their ability to think reflectively in and on their work. Diversity in experiences seems to have little impact on the level of mathematics teacher's reflective thinking.

In their answers to the first open-ended question, mathematics teachers did not realize that reflective thinking is considered an important learning outcome that should be taken care of in mathematics learning. In this regard, [46], stressed that reflective thinking involves "consideration of the larger context, the meaning, and the implications of an experience or action. Reflective thinking, shares some of the wider skills such as analysing and making judgments which are related to critical thinking. [14] defined reflective thinking as "an active, persistent, and careful consideration of a belief or supposed form of knowledge, of the grounds that support that knowledge, and the further conclusions to which that knowledge leads" (p. 118). [47] stressed that reflective thinking is more similar to critical thinking because both types of thinking are directed thinking that focuses on a desired outcome. Successful math teachers are those who have clear knowledge of the content being taught and have the ability to control their teaching by actively involved in reflective thinking reflectively.

Mathematics teachers answered the second question. Some teachers were able to identify some characteristics of a teacher practicing reflective thinking. In this regard, literature provided some characteristics as well as teachers' reflective thinking practices. For example [48], indicated that teacher reflection can generally be characterized as having the ability to "retrospection, problem-solving, critical analysis, and putting thoughts into action" (p. 3). In addition [48, P. 9], suggested the following characteristics: a) reflection as retrospective analysis, b) reflection as problem-solving, c) critical reflection of self, and d) reflection on beliefs about self. [15] added that reflective teachers should be concerned about reflection-in-action and reflection-on-action. This means that, in addition to reflecting ahead of teaching or during the preparation of teaching, the teacher should also reflect while teaching. In this regard [49],

indicated that teachers should have the ability to think critically about their assumptions and actions. [48, 50], described the reflective teacher there as having four learning processes: "assumption analysis, contextual awareness, imaginative speculation, and reflective scepticism".

The answers to the three open-ended questions indicated that the participants did not know enough about reflective thinking and its characteristics, nor did they know about its importance in their teaching profession. The teachers failed to accurately describe the characteristics of a reflective teacher. Certain studies [30, 51, 52] have argued that reflective practices play an important role in facilitating teaching, learning, and understanding, and thus should be part of teachers' professional development. Teachers' reflective thinking about, and as part of, their teaching practices may help them improve their practices and their students' abilities. Reflective thinking and practice share the following skills:

- Self-questioning

- Examination of their own practices to plan for improvement

- Collaboration with other professionals in order to improve

- Consideration of children's perspectives

- Eagerness to gain more perspectives

- Attention to detail

- Examination of the environment

- Full engagement in argumentation about and in their work

Reflective thinking enables teachers as well as students to develop teaching and learning skills such as; questioning, ability to self-assess, awareness of how one learns, and gaining self-knowledge.

## Conclusions and implications

The researchers conducted this research to shed light on the effectiveness of reflective thinking in the development of mathematics teachers. This study attempted to prove that reflective thinking is a useful means for mathematics teachers to explore themselves and to enhance their professional development. Therefore, the study used a validated Reflective Thinking Teacher Questionnaire (RTTQ) to assess the reflective thinking of mathematics teachers while teaching. The developer of the RTTQ has argued that this instrument is capable of providing valuable information regarding the levels of teachers' reflective thinking about their profession. The study also used a short open-ended to explore more information about mathematics teachers' views of reflective thinking, as well as their levels of implementing such a type of thinking in their practices. The results for all the themes concerning math teachers' reflective thinking examined in this study indicated that they fluctuated around the introductory, intermediate, and advanced levels. This result shows that teachers are critically in need of appropriate skills' development. The findings of this study suggest that self-reflective practices constitute an important aspect of teachers' positive involvement in school—and in their professional lives, in general. The results implied that teachers, Continuing Professional Development providers, teacher educators, and researchers may benefit from the results gained from the application of the RTTQ. Teachers (especially novices) can use this tool to assess their current levels of reflective thinking and their subsequent development over time, as well as for the preparation of teacher development programs. Also, supervisors can use the RTTQ as a

diagnostic for teachers working within their area of responsibility. Teacher educators and CPD providers can use the questionnaire to assess the progress of pre-service teachers' reflective thinking, as well as in the design of their preparation programs. Therefore, it is interesting to know how teachers' reflective thinking abilities can influence teachers' performance in mathematics classroom practice.

In short, this study is important for teacher development programs. Its main goal was to examine math teachers' reflective thinking skills in the context of how they are applied to their understanding of these skills in their teaching practices. The adequate instruments and proper statistical analysis were performed appropriately and rigorously. We concluded that the teachers' practices concerning all the themes of reflective thinking addressed by the questionnaire, especially their ability to self-assess, fluctuated around the introductory, intermediate, and advanced levels. We expect to strengthen the rationale for motivating further research exploring Saudi Arabian math teachers'. As a result, this study focused on a topic that is relevant and of general interest to readers in science education around the world.

## Acknowledgments

We thank the Deanship of Scientific Research, and the excellent center of Science and Mathematics Education, King Saud University, for their support in conducting this research

## Author Contributions

**Data curation:** Majed A. Naji.

**Formal analysis:** Majed A. Naji.

**Funding acquisition:** Abdulwali H. Aldahmash.

**Methodology:** Abdulwali H. Aldahmash.

**Resources:** Majed A. Naji.

**Writing – review & editing:** Abdulwali H. Aldahmash, Samar Ab. Alshalhoub.

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
