## [Decision Letter · Decision Letter 0]

20 Jul 2021

PONE-D-21-16098

Mathematics teachers’ reflective thinking: Level of understanding and implementation in their professional practices

PLOS ONE

Dear Dr. Aldahmash, 

Thank you for submitting your manuscript to PLOS ONE. After careful consideration, we feel that it has merit but does not fully meet PLOS ONE’s publication criteria as it currently stands. Therefore, we invite you to submit a revised version of the manuscript that addresses the points raised during the review process.

We look forward to receiving your revised manuscript.

Kind regards,

José Gutiérrez-Pérez

Academic Editor

PLOS ONE

"We thank the Excellent Research Center of Science and Mathematics Education, ECSME for

their support in conducting this research"

"This research was supported by the Deanship of Scientific Research, Research at King Saud University, Group no. (RG-1440-123) "

**Comments to the Author**

1. Is the manuscript technically sound, and do the data support the conclusions?

Partly

2. Has the statistical analysis been performed appropriately and rigorously? 

Parthly

3. Have the authors made all data underlying the findings in their manuscript fully available?

Patly

4. Is the manuscript presented in an intelligible fashion and written in standard English?

Partly

5. Review Comments to the Author

Reviewer #1: Measuring teachers’ reflective thinking on teaching practices during mathematics classroom is important to the teacher development programs. This study’s main goal is to examine the math teachers’ skills in the context of how they are applied to their understanding of their teaching practices. The adequate instrument and proper statistical analysis were performed appropriately and rigorously. The authors conluded that the teachers’ practices concerning all the themes of reflective thinking addressed by the questionnaire, especially their ability to self-assess, fluctuated around the introductory, intermediate, and advanced levels. The authors expect to strengthen the rationale motivating further research exploring Saudi Arabian math teachers’. Because of this, the current study is on a topicof relevance and general interest to the readers of the journal. It is suggested to publish. Please carefully proof-read spell check to eliminate typing and grammatical errors before its acceptance.

Reviewer #2: The paper lacks a lot of important details on teaching mathematics, why is it important for mathematics teachers conduct reflective thinking in teaching mathematics. The literature review was done in a brief manner. It is suggested the authors focus on addressing the literature and theoretical gaps before presenting the problem statement.

There was no discussion of the instruments available to measure mathematics teachers reflective thinking and why the 1997's instrument was used in this study.

Initially I thought the research only collected quantitative data as there was no explanation of qualitative data collection in the research method. Suddenly, there was an interview data after the quantitative data was presented. I am not sure the research design of this study. There was also no discussion on how the interview data was analysed.

The discussion was not linked to the literature review of the paper hence lacks of significant contribution towards the body of knowledge in teacher development.

There are also many unforgiven grammatical and spelling mistakes.

Hence, it is proposed that this paper to be either given a major revision or rejected from publication in a prestigious journal as PLOS.

Reviewer #3: Here are my comments to the paper:

1. The first comment is that there are a lot of citations whose sources are not in the reference list:

- Duban, Yanpar, & Yelken, 2010

- Dymoke and Harrison (2008)

- (Cohen et al., 2000)

- Branch & Paranjape (2002)

- Halpern (1996)

- Boody (2008)

- Riddell (2007)

- Brookfield (1988)

- Shermis (1999)

2. Check the following references which are not cited in the text:

-Chen, D., & Stroup, W. (1993). General system theory: Toward a conceptual framework for

science and technology education for all. Journal of Science Education and

Technology, 2(3), 447-459.

-Bell, P., & Linn, M. C. (2000). Scientific arguments as learning artifacts: Designing for

learning from the web with KIE. International Journal of Science Education, 22(8), 797–

817. https://doi.org/10.1080/095006900412284

- Hood, S. (2008). Linguistics and education: An International Research Journal, 19(4), 351–

365.

3. Theoretically and methodologically the work is well founded.

4. My main concerns are in the discussion:

- How do authors justify the predominance of the level "Introductory, intermediate and advanced"in the light of Tables 1, 2 and 3. Please elaborate more on this issue.

- On paragraph "Concerning the third theme,..., (Saavedra & Opfer, 2012), there is a repetition of categories related to the second theme.

5. Check out the following mistakes in the text:

- (paragraph Dewey (1933) and Schon (1983)...): "Reflection represents the highest level of the eight categories of argumentation", Which are these eight categories?

- (Ennis, 1996; 1987) (Ennis, 1987, 1996)

- Related studies have been don  Related studies have been done

- For example, Fariba Mirzaeia, Fatin Aliah, ...-> Mirzaei, Phang and Kashefi (2014)

- Choy, & Oo, (2012) Choy and Oo (2012)

- "teachers neglect the practice these four" "Teachers neglect the practice of these four..."

- Oo, Pou & Choy, Chee (2010) Oo and Choy (2010)

- Fariba et al (2013)->Mirzaei et al. (2014)

- (Fernandez, Karsenti & Charlin, 2014)-> (Nguyen et al., 2014)

- School mathematics teachers on-duty  school mathematics teachers on duty.

- Strongly agree: From 4:21 to 5:00 From 4.21 to 5.00, check this throughout the text!!!

- "were made to translation of the statements" were made to the translation of the statements

- "Cronbach's Alpha's coefficient-> Cronbach's Alpha Coefficient

- "ability to self-asses" ability to self-assess

- Schon (1987) added 1983 or 1987, check the year of the reference.

- (Allinder, 1994) (Allinder, 2009), check the year of the reference.

- Explain what PD programs and CPD providers stand for.

Reviewer #4: This paper reports the results of a large survey about mathematics teachers’ reflective thinking, where the questionnaire used was translated from English to Arabic and used in Saudi Arabia. I think that the outcomes might be of interest to a practitioner journal in that region. But, as presented here, the work does not make a contribution to knowledge at the level that would be expected in an international research journal. Below, I explain why, and also add some minor notes that the authors might wish to take into account if they decide to resubmit to this.

The main issue I see is that the paper’s main results are simply the average scores across the participants for each of the 33 self-report questionnaire items. These, on their own, do not tell us much. They are not really comparable with one another – what would it mean to say that teachers are more or less inclined to observe their performance than they are aware of how one learns? And, here, they are not related to anything else – for instance, we do not find out whether more experienced teachers tend to be more reflective (or to score higher on some subscale), or whether mathematics teachers are more or less reflective than teachers in other subjects, or whether more reflective teachers get better results or more participation from their students. Perhaps the authors have the data to link the scores to another construct like one of these. If so, that would strengthen the paper considerably.

To list some more minor issues:

• Some aspects of the introductory sections use strong language (“cannot”, “essential” etc.), which sound like the authors are making an absolute judgement about what is right or necessary rather than relying on research evidence.

• There is more than one list of constructs related to reflective thinking (Choy & Oo, Fariba et al.) but the authors do not explain whether and how these are linked.

• For an international journal, some information on the Saudi Arabian context will be needed, e.g. what age are students in high school, do all teachers teach in the same types of school, etc. This needn’t be long, but would help international readers to compare with their own contexts.

• The abstract says that the survey involved 602 teachers, but apparently only 322 responded.

• There are some numbers unnecessarily in brackets in the description of the instrument.

• Abbreviations and capitalisation are used inconsistently, e.g. RTTW vs. RTATQ, capitalising Excellent Centre or not.

• In reading about the scale, I was initially confused by the referencing and thought Williams & Burden had constructed it.

• I thought the ranges (strongly agree from 4.21 to 5.00 etc.) confusing and unnecessary. If I know there is a scale from 1 to 5, I can interpret what an average of 4.6 means. Also, I think that “weighted average” is misused here, and the authors just mean “mean”.

• The authors say that their main goal was to discover whether there was a statistically significant difference in mathematics teachers’ practices of reflective thinking, but they do not report any statistical tests about differences so this is not addressed.

• I was confused by the fact that the I, In and A items appear in different orders in different sections of the table. This makes it harder to compare across them.

• There is no report of the sampling strategy or analysis process for the qualitative interviews, and these are not systematically related to the quantitative data.

---

## [Author Response · Author response to Decision Letter 0]

4 Aug 2021

Rebuttal letter

PLOS ONE Decision: Revision required [PONE-D-21-16098] - [EMID:439b9ab96b8daaf1]‏‏ 

Dear José Gutiérrez-Pérez

Academic Editor

PLOS ONE

I hope you are doing very well

Here is a rebuttal letter including responses to the editor and the review’s comments. I have taken care of almost all comment and concerns raised by the reviewers and the editor. All responses are highlighted in green color. 

Dear Dr. Aldahmash, 

Thank you for submitting your manuscript to PLOS ONE. After careful consideration, we feel that it has merit but does not fully meet PLOS ONE’s publication criteria as it currently stands. Therefore, we invite you to submit a revised version of the manuscript that addresses the points raised during the review process.

Reply: Regarding the financial status, I will pay the fee from my own budget. However, if the fees are high, I hope that a percent of the due money could be waved due to the fact that the fund is not high enough and that the first and third authors are Yemeni nationals, while the second author is a Saudi national. 

Reply: I think these protocols do not apply to our article, because it is an educational type research. 

We look forward to receiving your revised manuscript.

Kind regards,

José Gutiérrez-Pérez

Academic Editor

PLOS ONE

Reply: requirements were followed.

Reply: the ethical committee sent us an email regarding their consent to apply the reflective thinking questionnaire on mathematics teachers. 

I have included the following statements:

• The participants in this study sample were informed of the study objectives at the beginning, in the instruction part, of the questionnaire. They were also informed that participation was not obligatory, and that they did not have to write their names or identity. In addition, this study had been approved by the Research Ethics Committee of Deanship of Scientific Research, King Saud University.

• We obtained informed written consent from the Research Ethics Committee of Deanship of Scientific Research, King Saud University via email

• I ensure that all the authors mentioned in the manuscript have agreed for authorship, read and approved the manuscript, and given consent for submission and subsequent publication of the manuscript.

 Reply: Not applicable

"We thank the Excellent Research Center of Science and Mathematics Education, ECSME for

their support in conducting this research"

Reply: Ok

"This research was supported by the Deanship of Scientific Research, Research at King Saud University, Group no. (RG-1440-123) "

Reply: OK

Reply: I am and the third author are Yemen nationals, while the second author is a Saudi national. Therefore, I wish if you could reduce the open publishing fee. 

Comments to the Author

1. Is the manuscript technically sound, and do the data support the conclusions?

Partly

2. Has the statistical analysis been performed appropriately and rigorously?

Parthly

3. Have the authors made all data underlying the findings in their manuscript fully available?

Reply: OK

Patly

4. Is the manuscript presented in an intelligible fashion and written in standard English?

Partly

5. Review Comments to the Author

Reviewer #1: Measuring teachers’ reflective thinking on teaching practices during mathematics classroom is important to the teacher development programs. This study’s main goal is to examine the math teachers’ skills in the context of how they are applied to their understanding of their teaching practices. The adequate instrument and proper statistical analysis were performed appropriately and rigorously. The authors concluded that the teachers’ practices concerning all the themes of reflective thinking addressed by the questionnaire, especially their ability to self-assess, fluctuated around the introductory, intermediate, and advanced levels. The authors expect to strengthen the rationale motivating further research exploring Saudi Arabian math teachers’. Because of this, the current study is on a topic of relevance and general interest to the readers of the journal. It is suggested to publish. Please carefully proof-read spell check to eliminate typing and grammatical errors before its acceptance.

Reply: I have taken care of all comments and concerns raised by the first reviewer. 

Reviewer #2: The paper lacks a lot of important details on teaching mathematics, why is it important for mathematics teachers conduct reflective thinking in teaching mathematics. The literature review was done in a brief manner. It is suggested the authors focus on addressing the literature and theoretical gaps before presenting the problem statement.

Reply: I have added paragraphs to explain why is it important for mathematics teachers conduct reflective thinking in teaching mathematics. (p.3 )

There was no discussion of the instruments available to measure mathematics teachers reflective thinking and why the 1997's instrument was used in this study.

Reply: I have added a paragraph discussing the instruments available to measure reflective thinking among teachers in general and mathematics teachers in particular, and explaining why the 1997's instrument was used in this study. (p.6)

Initially I thought the research only collected quantitative data as there was no explanation of qualitative data collection in the research method. Suddenly, there was an interview data after the quantitative data was presented. I am not sure the research design of this study. There was also no discussion on how the interview data was analysed.

Reply: an explanation of qualitative data collection was added in the research method (p.8)

The discussion was not linked to the literature review of the paper hence lacks of significant contribution towards the body of knowledge in teacher development.

Reply: discussion was linked to the literature review to ensure the significant contribution towards the body of knowledge in teacher development (p. 19-20)

There are also many unforgiven grammatical and spelling mistakes.

Reply: grammatical and spelling check has been made throughout the manuscript.

Hence, it is proposed that this paper to be either given a major revision or rejected from publication in a prestigious journal as PLOS.

Reply: Major revision has been implemented.

Reviewer #3: Here are my comments to the paper:

1. The first comment is that there are a lot of citations whose sources are not in the reference list:

- Duban, Yanpar, & Yelken, 2010

- Dymoke and Harrison (2008)

- (Cohen et al., 2000)

- Branch & Paranjape (2002)

- Halpern (1996)

- Boody (2008)

- Riddell (2007)

- Brookfield (1988)

- Shermis (1999)

Reply: All references have been carefully checked whether in the text of the in the list (see the tracked version of the manuscript)

2. Check the following references which are not cited in the text:

-Chen, D., & Stroup, W. (1993). General system theory: Toward a conceptual framework for

science and technology education for all. Journal of Science Education and

Technology, 2(3), 447-459.

-Bell, P., & Linn, M. C. (2000). Scientific arguments as learning artifacts: Designing for

learning from the web with KIE. International Journal of Science Education, 22(8), 797–

817. https://doi.org/10.1080/095006900412284

- Hood, S. (2008). Linguistics and education: An International Research Journal, 19(4), 351–

365.

Reply: All references have been carefully checked whether in the text of the in the list (see the tracked version of the manuscript)

3. Theoretically and methodologically the work is well founded.

Reply: Don

4. My main concerns are in the discussion:

- How do authors justify the predominance of the level "Introductory, intermediate and advanced"in the light of Tables 1, 2 and 3. Please elaborate more on this issue.

- On paragraph "Concerning the third theme,..., (Saavedra & Opfer, 2012), there is a repetition of categories related to the second theme.

Reply: I have made the correction of this part (P.13)

5. Check out the following mistakes in the text:

- (paragraph Dewey (1933) and Schon (1983)...): "Reflection represents the highest level of the eight categories of argumentation", Which are these eight categories?

- (Ennis, 1996; 1987) (Ennis, 1987, 1996)

- Related studies have been don  Related studies have been done

- For example, Fariba Mirzaeia, Fatin Aliah, ...-> Mirzaei, Phang and Kashefi (2014)

- Choy, & Oo, (2012) Choy and Oo (2012)

- "teachers neglect the practice these four" "Teachers neglect the practice of these four..."

- Oo, Pou & Choy, Chee (2010) Oo and Choy (2010)

- Fariba et al (2013)->Mirzaei et al. (2014)

- (Fernandez, Karsenti & Charlin, 2014)-> (Nguyen et al., 2014)

- School mathematics teachers on-duty  school mathematics teachers on duty.

- Strongly agree: From 4:21 to 5:00 From 4.21 to 5.00, check this throughout the text!!!

- "were made to translation of the statements" were made to the translation of the statements

- "Cronbach's Alpha's coefficient-> Cronbach's Alpha Coefficient

- "ability to self-asses" ability to self-assess

- Schon (1987) added 1983 or 1987, check the year of the reference.

- (Allinder, 1994) (Allinder, 2009), check the year of the reference.

- Explain what PD programs and CPD providers stand for.

Reply: I have made the required correction of all mistakes. (all citations have been changed according to the journal system) 

Reviewer #4: This paper reports the results of a large survey about mathematics teachers’ reflective thinking, where the questionnaire used was translated from English to Arabic and used in Saudi Arabia. I think that the outcomes might be of interest to a practitioner journal in that region. But, as presented here, the work does not make a contribution to knowledge at the level that would be expected in an international research journal. Below, I explain why, and also add some minor notes that the authors might wish to take into account if they decide to resubmit to this.

Reply: I have taken care of most of the concerns raised by the 4rth reviewer.

The main issue I see is that the paper’s main results are simply the average scores across the participants for each of the 33 self-report questionnaire items. These, on their own, do not tell us much. They are not really comparable with one another – what would it mean to say that teachers are more or less inclined to observe their performance than they are aware of how one learns? And, here, they are not related to anything else – for instance, we do not find out whether more experienced teachers tend to be more reflective (or to score higher on some subscale), or whether mathematics teachers are more or less reflective than teachers in other subjects, or whether more reflective teachers get better results or more participation from their students. Perhaps the authors have the data to link the scores to another construct like one of these. If so, that would strengthen the paper considerably.

Reply: an ANOVA test was done to see if there are differences in teachers’ reflective thinking related to experiences, in addition to another table that sum-up the results in a more clearer way (see p. 14-15)

To list some more minor issues:

• Some aspects of the introductory sections use strong language (“cannot”, “essential” etc.), which sound like the authors are making an absolute judgement about what is right or necessary rather than relying on research evidence.

Reply: I have changed many wordings to avoid making an absolute judgement.

• There is more than one list of constructs related to reflective thinking (Choy & Oo, Fariba et al.) but the authors do not explain whether and how these are linked.

Reply: I have taken care of this issue 

• For an international journal, some information on the Saudi Arabian context will be needed, e.g. what age are students in high school, do all teachers teach in the same types of school, etc. This needn’t be long, but would help international readers to compare with their own contexts.

Reply: age of students and characteristics of schools have been added (p.6)

• The abstract says that the survey involved 602 teachers, but apparently only 322 responded.

Reply: sample # have been corrected

• There are some numbers unnecessarily in brackets in the description of the instrument.

Reply: Numbers are necessary in the description of the instruments and their items (p.6 )

• Abbreviations and capitalisation are used inconsistently, e.g. RTTW vs. RTATQ, capitalising Excellent Centre or not.

Reply: I capitalized all

• In reading about the scale, I was initially confused by the referencing and thought Williams & Burden had constructed it.

Reply: Williams & Burden had constructed the instruments and was modified by Choy & Oo (2012) (p.6 )

• I thought the ranges (strongly agree from 4.21 to 5.00 etc.) confusing and unnecessary. If I know there is a scale from 1 to 5, I can interpret what an average of 4.6 means. Also, I think that “weighted average” is misused here, and the authors just mean “mean”.

Reply: the weighted averages have been replaced by means 

• The authors say that their main goal was to discover whether there was a statistically significant difference in mathematics teachers’ practices of reflective thinking, but they do not report any statistical tests about differences so this is not addressed.

Reply: I have added table 5 to present statistical differences according to experiences (p. 15)

• I was confused by the fact that the I, In and A items appear in different orders in different sections of the table. This makes it harder to compare across them.

Reply: I have changed the orders (p 9-10) 

• There is no report of the sampling strategy or analysis process for the qualitative interviews, and these are not systematically related to the quantitative da

Reply: the open ended question were given to all samples, because they were combined with the survey (p.6 )

---

## [Decision Letter · Decision Letter 1]

3 Sep 2021

PONE-D-21-16098R1Mathematics teachers’ reflective thinking: Level of understanding and implementation in their professional practicesPLOS ONE

Dear Dr. Aldahmash,

Thank you for submitting your manuscript to PLOS ONE. After careful consideration, we feel that it has merit but does not fully meet PLOS ONE’s publication criteria as it currently stands. Therefore, we invite you to submit a revised version of the manuscript that addresses the points raised during the review process.

We look forward to receiving your revised manuscript.

Kind regards,

José Gutiérrez-Pérez

Academic Editor

PLOS ONE

Journal Requirements:

**Comments to the Author**

In general, I observe an improvement of the paper. My main concern is related to the interpretation of Table 1, 2, 3 and 4.

Note that in table 1, item #1, the weighted mean is 3.02, rather than 3.10

First, in the following paragraph:

The teachers’ ability to "make judgements" was most prevalent at the introductory level, where the

highest weighted average was 4.42, indicating that the introductory-level teachers "strongly agreed"

with this item.

The value 4.42 is related to the advanced level, not the introductory level as authors stated.

Second, in the following paragraph:

In short, math teachers’ ability to "observe their performance" was most prevalent at the

introductory level, their ability to "use feedback and evidence" was most prevalent at both the

intermediate and advanced levels, their ability to "find and analyze patterns" was most prevalent at the

advanced level, and their ability to "make judgments" was most prevalent at the introductory level. As

a result, math teachers’ ability to "self-assess" was found predominantly at the intermediate level, where

the value for the weighted average was 3.85.

Check that the highest weighted mean in "observing the performance" is 4.60 which corresponds to Advanced level, and in "make judgements" is 4.42 which corresponds to Advanced level

Where does the value 3.85 come from? I think that authors made the average of each level (I, In and A) and then chosen he highest value:

- For example, from table 1. Level I has an average of (3.02+2.78+4.39+2.75)/4=3.24 ; Level In has an average of 3.98; and Level A has an average of 4.37

So the proper value is 3.98

Third, in the following paragraph:

This theme consisted of three sub-themes, as shown in Table 2. The results for the first sub-theme,

"concepts and misconceptions," indicated that math teachers’ awareness of concepts and

misconceptions was most prominently observed between the intermediate and advanced levels, where

the mean for the frequencies were 4.14 and 4.28, respectively. Meanwhile, the results of the second

sub-theme, "knowledge construction," indicated that the teachers’ ability to construct knowledge also

concentrated around the intermediate and advanced levels. Similarly, the results for the third sub-theme,

"metacognition," showed that teachers’ belief in their metacognitive ability was most prevalent at the

intermediate and advanced levels; the intermediate weighted average for the frequency of this level was

4.28.

Where does value 4.28 come from? By calculating the average of each level (N=3, sub-themes), Introductory level has an average of 2.81; Intermediate level (4.286..., so 4.29); Advanced level (4.37).

As done before in table 1, authors may conclude that 4.37 is that is an advanced level predominance in theme "teachers' awareness of how one learns"

So in paragraph:

The means for the categories of the theme, "awareness of how one learns," indicated that teachers’

abilities to identify "concepts and misconceptions," "knowledge construction," and "metacognition" are

predominantly at the [[[intermediate level]]. The proper level is advanced (4.37)

Related to table 4, I do not understand the statement and ask the authors to explain a little more. "All the values of the means were found to center around the "agree" response, which the grading rules indicated was between 3.41 and 4.2" (3:41 and 4:2 are typos).

The important thing is that authors should keep the same way to discuss the results from the four themes.

In table 5, Sub total cells includes the average of each level for each skill. The total cells are the average of the four sub total values. This has to be explicited.

Finally, I have detected a mistake in the "Discussion" section (Studies [49, 9; 4], It should be Studies [4, 9, 49]

Congratulations to authors.

---

## [Author Response · Author response to Decision Letter 1]

10 Sep 2021

Rebuttal Litter2

Journal Requirements:

Answer:

Most of the references had been reviewed and reordered.

The retracted references were not cited in the text body, and were irrelevant to the paper. 

all references were reordered according to their original sources to ensure that it is complete and correct (see the markup copy in which each reference number was aligned with its original source and was written as a merging comment. (P. 17-19)

Comments to the Author

In general, I observe an improvement of the paper. My main concern is related to the interpretation of Table 1, 2, 3 and 4.

Note that in table 1, item #1, the weighted mean is 3.02, rather than 3.10

Answer: 

weighted mean in table1 had been changed from 3.10 to 3.02

First, in the following paragraph:

The teachers’ ability to "make judgements" was most prevalent at the introductory level, where the highest weighted average was 4.42, indicating that the introductory-level teachers "strongly agreed"

with this item.

The value 4.42 is related to the advanced level, not the introductory level as authors stated.

Answer:

• According to the highest mean (4.4), the level of the sub-theme “make judgements“ was changed to advanced rather than intermediated or introductory (Table1).

• 

According to the highest mean (4.60) the level of the sub-theme “ability to observe their performance" was changed to advanced rather than intermediated (Table1). introductory

• According to the highest mean (4.04), the level of the sub-theme “ability to use feedback and evidence " “was changed to intermediated (Table1).

• According to the highest mean(4.41), the level of the sub-theme “ability to Finding and analyzing patterns " “ was changed to advanced (Table1).

• The level for the hole theme “ ability to self-assess is Advanced because the highest mean was 4.37, whereas the mean of the intermediate level is 4.29, and the introductory level was 2.81

Second, in the following paragraph:

In short, math teachers’ ability to "observe their performance" was most prevalent at the

introductory level, their ability to "use feedback and evidence" was most prevalent at both the intermediate and advanced levels, their ability to "find and analyze patterns" was most prevalent at the advanced level, and their ability to "make judgments" was most prevalent at the introductory level. As a result, math teachers’ ability to "self-assess" was found predominantly at the intermediate level, where

the value for the weighted average was 3.85.

Check that the highest weighted mean in "observing the performance" is 4.60 which corresponds to Advanced level, and in "make judgements" is 4.42 which corresponds to Advanced level, Where does the value 3.85 come from? I think that authors made the average of each level (I, In and A) and then chosen he highest value:

- For example, from table 1. Level I has an average of (3.02+2.78+4.39+2.75)/4=3.24 ; Level In has an average of 3.98; and Level A has an average of 4.37

So the proper value is 3.98 

Answer:

The level of the sub-theme “ability to observe their performance" “was changed to advanced rather than intermediated or introductory

The level of the sub-theme “ability to use feedback and evidence " “was changed to advanced rather than intermediated or introductory

The total mean for the intermediated level of the first theme was changed to 3.98

The level for the hole theme is Advanced because the highest mean was 4.37, whereas the mean of the intermediate level is 4.29, and the introductory level was 2.81 (P. 6-7)

The total of the averages for each level were calculated and placed under the table

Third, in the following paragraph:

This theme consisted of three sub-themes, as shown in Table 2. The results for the first sub-theme, "concepts and misconceptions," indicated that math teachers’ awareness of concepts and misconceptions was most prominently observed between the intermediate and advanced levels, where the mean for the frequencies were 4.14 and 4.28, respectively. Meanwhile, the results of the second sub-theme, "knowledge construction," indicated that the teachers’ ability to construct knowledge also concentrated around the intermediate and advanced levels. Similarly, the results for the third sub-theme, "metacognition," showed that teachers’ belief in their metacognitive ability was most prevalent at the intermediate and advanced levels; the intermediate weighted average for the frequency of this level was

4.28.

Where does value 4.28 come from? By calculating the average of each level (N=3, sub-themes), Introductory level has an average of 2.81; Intermediate level (4.286..., so 4.29); Advanced level (4.37). As done before in table 1, authors may conclude that 4.37 is that is an advanced level predominance in theme "teachers' awareness of how one learns" (p. 7-8)

So in paragraph:

The means for the categories of the theme, "awareness of how one learns," indicated that teachers’ abilities to identify "concepts and misconceptions," "knowledge construction," and "metacognition" are predominantly at the [[[intermediate level]]. The proper level is advanced (4.37).

Answer:

The proper level was changed to advanced (4.37) (p. 7-8)

Related to table 4, I do not understand the statement and ask the authors to explain a little more. "All the values of the means were found to center around the "agree" response, which the grading rules indicated was between 3.41 and 4.2" (3:41 and 4:2 are typos).

Answer:

The level of the sub-theme “Concepts and misconceptions “ is Advanced because the highest mean value is 4.28 

The level of the sub-theme “Knowledge construction “ is Advanced because the highest mean value is 4.48

The level of the sub-theme “Metacognition “ is Intermediate because the highest mean value is 4.41

The level for the hole theme is Advanced because the highest mean was 4.37, whereas the mean of the intermediate level is 4.29, and the introductory level was 2.81 . (p. 10-11)

The total of the averages for each level were calculated and placed under the table

In table 5, Subtotal cells includes the average of each level for each skill. The total cells are the average of the four subtotal values. This has to be explicited.

Answer:

Percentages have been added to table5 to make the interpretation clearer, and to support the results reported earlier. In addition, some changes were made to the interpretation of the data to be explicated. (p. 11)

The total of the averages for each level were calculated and placed under the table

The important thing is that authors should keep the same way to discuss the results from the four themes.

Finally, I have detected a mistake in the "Discussion" section (Studies [49, 9; 4], It should be Studies [4, 9, 49]

Answer:

Parts of the discussion has been change to be aligned with the changes made to the results parts (p. 13-15)

The order of the studies [49, 9; 4] has been changed to [4, 9, 49]. (p. 13-15)

Congratulations to authors.

Great thanks to the reviewers, especially who made the second revision, who made thorough reading, and was very keen in his revision. 

---

## [Editor Report · Decision Letter 2]

20 Sep 2021

Mathematics teachers’ reflective thinking: Level of understanding and implementation in their professional practices

PONE-D-21-16098R2

Dear Dr. Aldahmash,

We’re pleased to inform you that your manuscript has been judged scientifically suitable for publication and will be formally accepted for publication once it meets all outstanding technical requirements.

Kind regards,

José Gutiérrez-Pérez

Academic Editor

PLOS ONE

---

## [Editor Report · Acceptance letter]

27 Sep 2021

PONE-D-21-16098R2 

Mathematics teachers’ reflective thinking: Level of understanding and implementation in their professional practices 

Dear Dr. Aldahmash:

I'm pleased to inform you that your manuscript has been deemed suitable for publication in PLOS ONE. Congratulations! Your manuscript is now with our production department. 

Kind regards, 

on behalf of

Dr. José Gutiérrez-Pérez 

Academic Editor

PLOS ONE